# Detection of Asymptomatic Mpox Carriers among High-Ri Men Who Have Sex with Men: A Prospective Analysis

**DOI:** 10.3390/pathogens12060798

**Published:** 2023-06-03

**Authors:** Roberto Rossotti, Daniele Calzavara, Massimo Cernuschi, Federico D’Amico, Anna De Bona, Roberto Repossi, Davide Moschese, Simona Bossolasco, Alessandro Tavelli, Camilla Muccini, Giovanni Mulé, Antonella d’Arminio Monforte

**Affiliations:** 1Department of Infectious Diseases, ASST Grande Ospedale Metropolitano Niguarda, 20162 Milan, Italy; 2Milano Checkpoint, 20124 Milan, Italy; 3Department of Infectious Diseases, San Raffaele Scientific Institute, 20127 Milan, Italy; 4Department of Infectious Diseases, ASST Santi Paolo e Carlo, 20142 Milan, Italy; 5I Division of Infectious Diseases, ASST Fatebenefratelli-Sacco, 20157 Milan, Italy; 6ICONA Foundation, 20142 Milan, Italy

**Keywords:** mpox, asymptomatic infection, point-of-care testing, PrEP

## Abstract

Mpox is traditionally considered a zoonotic disease with endemic circulation in Africa, but the 2022–2023 outbreak reached an unprecedented high number of cases in non-endemic countries, so that it was declared a public health emergency of international concern. The reasons for this extensive global spread, characterized by sexual transmission amongst men who have sex with men (MSM), have not been fully clarified. The existence of asymptomatic carriers with viable viral shedding might be an explanation and is under-debated after retrospective studies suggested that infection without symptoms might have a prevalence of 6.5%. We aimed to prospectively assess the presence of mpox infection in asymptomatic high-risk MSM using HIV pre-exposure prophylaxis and living with HIV. We selected individuals with no signs of active infection nor suggestive symptoms in the previous 21 days. Eligible individuals collected oral and anal swabs to undergo point-of-care testing for mpox and completed a 21-days follow-up. Seventy-two individuals were enrolled, and none tested positive for mpox infection nor developed symptoms during follow-up. We selected a high-risk population with a significant history of sexual exposure, but we failed to detect any asymptomatic infection. This observation might have important consequences in terms of contact management and epidemic control.

## 1. Introduction

Mpox (formerly known as monkeypox) is traditionally considered a zoonotic disease with endemic circulation in Africa [1]. However, from 2022–2023, a large outbreak occurred in non-endemic countries with peculiar clinical and epidemiological features characterized by a predominant sexual transmission observed in men who have sex with men (MSM) [2]. Conventional mpox manifestations are generally self-limiting, with fatality rates ranging from 1 to 10%, according to the viral clade [1]. Complications include pneumonitis, encephalitis, keratitis, and secondary bacterial infections [1]. Clinical picture typically begins with fever, followed by the development of multiple papular, vesicular-pustular, and ulcerative lesions on the face and body and prominent lymphadenopathy. A peculiar feature of the 2022–2023 outbreak is the regular involvement of sexually exposed mucosae [2]. The incubation period is customarily believed to be of 12 days (ranging from 5–21 days) [3]. However, another unexpected feature of the current epidemic is the shorter incubation, which seems to be reduced to 7 days [4]. Viral transmission to women is much less common, but the clinical features of those described in men include the presence of anal and genital lesions with prominent mucosal involvement, thus reflecting spread through sexual exposure [5]. Only 1.2% of cases are registered in individuals younger than 18 years, and less than 0.5% in individuals younger than 5 years. Three-quarters of pediatric cases have been reported from the WHO Americas region [6]. Clinical outcomes in children during the 2022–2023 outbreak appear milder compared to adults [7].

So far, few therapeutic strategies are available. Cidofovir and brincidofovir have shown in-vitro effectiveness against orthopoxviruses and some case series reported efficacy against mpox. However, the relevant kidney toxicity related to cidofovir, and liver toxicity related to brincidofovir, delivered substantial limitations to their use in clinical practice [8]. Tecovirimat was licensed in 2018 for smallpox and in 2022 for mpox [9]. A delayed viral infection onset, as well as a decrease in lesion formation reducing disease severity and mortality, has been reported [10]. Drug-resistance development is a major limitation for its use [11]. Tecovirimat has been evaluated in humans against mpox but lacks randomized trials [12]. No specific vaccine effective against mpox is available yet. Smallpox vaccines are employed since they exhibit at least 85% cross-protection [13,14]. Currently, only modified vaccinia Ankara (MVA–BN) is approved and available in Italy for laboratory personnel and for MSM with recent multiple sexual partners, those who have participated in group sex events, those who have attended gay clubs and cruising facilities, those who have had a recent STI diagnosis, and those who have engaged in chemsex practices [15]. Vaccination started in select Italian regions from mid-August 2022, but with very few doses. In the Lombardy region, where our site is located, roughly 6500 individuals have been vaccinated, a quantity largely inadequate to protect an estimated number above 100,000 of potentially eligible subjects (Welfare DG, Regione Lombardia, unpublished data). Unfortunately, some alarming reports suggest that infection could occur even after vaccination, with a prevalence higher than expected [16] and with potential severe illnesses [17].

On 23 July 2022, the World Health Organization (WHO) declared the ongoing epidemic of mpox a public health emergency of international concern [18]. As of April 2023, more than 87,000 cases have been identified across more than 110 countries with an overall mortality rate below 1% [19]. Nevertheless, lethality in immune-suppressed individuals could reach 25% [20]. As mentioned, cases have been registered predominantly in MSM, especially in people living with HIV (PLWH, above 40% of cases) and pre-exposure prophylaxis (PrEP) users (above 30% of cases) [2]. This novel trend, which mostly skewed toward sexual transmission amongst MSM, is in contrast with traditional mpox epidemiology [21].

The reasons for this unprecedented outbreak in non-endemic countries have not been clarified yet; virological, behavioral, and environmental factors might all play a role. Therefore, the possible existence of asymptomatic mpox carriers is under debate, even though asymptomatic cases are generally thought to have negligible weight in the spread of orthopoxvirus [22]. Historically, transmission was thought to be efficient only in the presence of active skin rash [23]. The isolation of the virus from throat cultures in asymptomatic people was believed to be of poor epidemiologic relevance [24]. Nevertheless, the 2022–2023 global outbreak and the novel human-to-human transmission evidence could provide proof that asymptomatic spread might indeed occur [25]. A retrospective Belgian study included 224 MSM and discovered three individuals (1.3%) with positive mpox tests without symptoms [26]. A second study from France retrospectively assessed anal swabs from MSM presenting for routine sexually transmitted infections (STI) screening. Excluding those who tested positive for gonorrhea and *Chlamydia trachomatis*, 13 out of 200 enrolled individuals tested positive for mpox (6.5%) [27]. These pivotal studies had some relevant limitations: their analysis design was exclusively retrospective. Not all the included subjects underwent concomitant full clinical evaluation; thus, the absence of symptoms was often presumptive. Therefore, strong conclusions on asymptomatic carriers could not be drawn.

Viral shedding without symptoms could substantially contribute to the transmission chain and might be addressed in public health policy to contain mpox. For instance, the WHO and the European Center for Disease Control and Prevention currently do not recommend the testing of close contacts of confirmed cases unless they present symptoms [28,29]. The evidence of asymptomatic contagious infections might require more quarantine disposal and additional testing that are not always widely available [30]. Data from prospective studies, including physical examination in exposed persons, are needed to achieve a greater knowledge of asymptomatic infections. The aims of the present study are: (i) Describing the prevalence of mpox infection in MSM PrEP users and PLWH without suggestive symptoms of orthopoxvirus contagion; (ii) Describing clinical and virological evolution, as well as the contagiousness of asymptomatic carriers.

## 2. Methods

A prospective monocentric analysis was designed and included individuals at major risk of viral exposure, such as PrEP users and PLWH. According to the regional law regulating mpox detection and management, testing might be performed only in two reference laboratories after an official disease notification to the health authority, forcing the notified subject to maintain quarantine until the availability of a negative result [31]. Thus, we decided to lead the study in Milano Checkpoint ETS, a community-based, peer-driven facility that provides sexual health care to the largest Italian cohort of PrEP users. Milano Checkpoint ETS was created in 2018 from the collaboration of several non-governmental organizations working in the field of HIV prevention and MSM health. In this out-of-hospital setting, STI diagnostics are routinely performed with point-of-care testing (POCT). Diagnosis confirmation, disease notification, and proper treatment are then provided by STI clinics present in the Milan metropolitan area that cooperate with Milano Checkpoint.

Study eligibility criteria were: (i) No fever, sore throat, chills, exhaustion, headache, myalgia, backache, or other constitutional disturbances in the previous 21 days; (ii) No lymphadenopathy in the previous 21 days; (iii) No skin lesions of novel appearance in the previous 21 days; (iv) No clinically evident symptoms present during a routine evaluation held by an infectious diseases specialist or by a trained peer; (v) No genital or anal pain not explained by other evident reasons; (vi) Not having received a diagnosis of mpox infection from May 2022. Having taken any dose of vaccination (including the full course during childhood) was not considered as an exclusion criterion.

PrEP users were consecutively evaluated during routine visits from September 2022. PLWH were recruited through advertisings on the official Milano Checkpoint social media informing about the chance of getting tested for mpox. After informed consent signature, all subjects filled a self-administered questionnaire on the RedCap platform collecting data on their sexual behaviors. Then, full clinical evaluation was performed, including careful inspection of skin throughout the whole-body surface; genital and perianal areas; the oral cavity; palms and soles; palpations of occipital, neck, submandibular, supraclavicular, axillary, retrocrural, inguinal, and popliteal lymph nodes. Endoscopic assessment of pharyngeal and anal cavities was not performed.

PrEP users were only screened for other STIs, as per routine clinical practice. Testing for HIV, HCV, and syphilis was performed with lateral-flow capillary qualitative POCTs. They self-collected urine, oral, and anal swabs to undergo gonorrhea and *Chlamydia trachomatis* testing through the rapid GeneXpert platform with Xpert CT/NG tests (Cepheid, Sunnyvale, CA, USA).

All enrolled individuals underwent mpox testing through the rapid Standard M10 platform with MPX/OPX test (SD Biosensor, Suwon, South Korea). This POCT recognizes different target genes, allowing for identification of the presence of orthopoxvirus to diagnose the presence of mpox, and then to distinguish between Clades I and II. MPX/OPX test was certified to be used on oral samples only. However, data from a Spanish cohort depicted more precocious and lower cycle threshold values with anal swabs when assessed with standard procedures [4]. We also decided to evaluate anal specimens in the analysis, although these were not included in the manufacturer’s instruction. Self-collection of pharyngeal and anal samples resulted in reliable and accurate detection of mpox [32]. Thus, eligible subjects self-collected additional oral and anal swabs to undergo mpox testing.

Each subject with a positive test would be directed to one of the clinical centers involved in mpox care to confirm the infection (ASST Grande Ospedale Metropolitano Niguarda, ASST Santi Paolo e Carlo). At least three samples would be collected (an EDTA blood sample, a urine sample, a swab taken from the oropharynx, and anal canal at the clinician’s discretion) to undergo standard PCR assays (RealStar^®^Orthopoxvirus PCR Kit 1.0, Altona Diagnostics, Hamburg, Germany). Cell-culture isolation would be used to assess replicating competence.

All enrolled asymptomatic individuals (either with a positive or a negative test) would be monitored for further symptom development in the following 21 days with a phone call every 3 days. In case of new onset of symptoms, a hospital clinical visit would be shortly arranged. Figure 1 summarizes study procedures.

Based on the available prevalence data of 6.5% in a population of 200 unselected individuals from the mentioned French study, a sample size of a minimum of 67 subjects (with a 95% confidence interval) was calculated. The planned time for enrollment was 12 weeks, starting from Institutional Review Board approval.

The study was conducted in accordance with the ethical standards of the Helsinki Declaration. Local Institutional Review Board approval was achieved (approval number S0025/2022). All patients signed a written informed consent to allow the use of their anonymized data and biological samples for clinical research.

Descriptive statistics (median and interquartile range [IQR] for continuous variables, absolute and relative [%] values for categorical variables) were used. Mann–Whitney U, for continuous variables, and Fisher’s exact test, for categorical variables, were applied to compare groups. Two-tailed *p*-values were calculated, and a value below 0.05 was considered statistically significant. Data management and analysis were performed using STATA package, Version 16.1 (College Station, TX, USA, StataCorp 2019).

## 3. Results

Institutional Review Board approval was delivered on 2 September 2022, when the outbreak was already in its declining phase. Although Milan alone represented more than 40% of all Italian cases, the city followed the national epidemiologic trend (Figure 2) [33]. Consequently, recruitment was prematurely discontinued on 20 October 2022 after only 7 out of the planned 12 weeks for screening since the number of cases detected was so low that proceeding was deemed as futile.

Screening involved 98 subjects: 25 were excluded because they did not meet the inclusion criteria (mainly because of a previous mpox diagnosis); one individual withdrew his consent for fear of quarantining policies in case of positivity. Seventy-two subjects underwent mpox screening: 63 PrEP users (87.5%) and 9 PLWH. They showed a median age of 37 (IQR 32–46) years and a high level of education (75.0% with a university degree). Sexual behavior suggested high-risk practices. They had a median of 11 (IQR 4–24) sexual intercourses in the previous 3 months; 13.9% had a recent STI diagnosis; 26.4% used recreational drugs. One-quarter of the study population achieved the full vaccine protection at the time of evaluation. Table 1 shows demographic, clinical, and behavioral features of the screened population. Enrolled subjects had a lower recreational drug use and a higher rate of vaccine uptake compared to those who failed screening.

Sixty-three PrEP users were screened for other STIs: two cases of syphilis (3.2%), two of gonorrhea (3.2%), and five *Chlamydia trachomatis* infections (7.9%) were detected. No case of HIV or HCV infection was found. None of the 72 enrolled subjects tested positive for mpox. During the 21 days of follow-up, nobody developed symptoms suggestive of mpox requiring further clinical evaluation.

## 4. Discussion

Our prospective study selected a population at considerable risk: the high screening failure rate suggests that we chose a group substantially exposed to the infection. Despite a significant sexual risk profile, we failed to detect any asymptomatic carrier at baseline, and nobody developed the disease during the 21-days of follow-up.

The unprecedented extent of transmission observed in the current outbreak hints that viral shedding by individuals who are asymptomatic—or mildly symptomatic—could play an important role. An estimation found that the proportion of undetected cases might be relevant, although with heterogeneity across countries [34]. In Italy, for instance, the estimated number of cases should be two times above the number of registered infections. A recent meta-analysis estimated a pooled percentage of asymptomatic infections of 9.1% [35]. A mathematical model predicted a basic R0 of 3.1 and a percentage of carriers without symptoms of 14% [36].

Data about asymptomatic orthopoxvirus infections are elusive. Sarkar performed cell cultures from throat samples of 328 asymptomatic family contacts of 52 smallpox patients. Variola virus grew in 34 contacts (10%): of them, only four subsequently developed smallpox [37]. The study was replicated in another small group of asymptomatic contacts of confirmed smallpox cases, finding again 10% positive cultures [38]. Concerning mpox, data from Africa in the 1980s provided evidence that approximately one-third of cases in unvaccinated individuals were subclinical [39].

The mentioned French and Belgian studies suffer from methodological limitations that could make it difficult to distinguish between truly asymptomatic subjects and individuals with mild symptoms that went unrecognized by patients and clinicians likewise [40]. The cited paper by Ferré and colleagues found that two out of 13 asymptomatic individuals later developed mpox. Furthermore, the same group by De Baetselier expanded its previous analysis, finding 14 additional cases [41]. Nine subjects were then found with symptoms not fulfilling mpox case definitions, while three individuals were pre-symptomatic and one had prodromal symptoms. A study performed during the MVA–BN vaccination campaign in Washington, DC, USA, enrolled 543 individuals, finding three positive samples (one pharyngeal and two anal swabs), showing an overall prevalence below 1% [42]. The virus results were viable in two cases. Not all the asymptomatic carriers had a consistent history of sexual exposure, and the only one with suggestive complaints had low quantification cycles (17.2), suggesting a possible correlation between viral load and symptom development. A study comparable to ours was performed in Hamburg, Germany, enrolling asymptomatic MSM without focusing on specific groups at risk. The authors screened 53 unselected individuals attending their STI Clinic, but they failed to detect any infection [43]. Another study from Milan found two asymptomatic MSM with detectable mpox genetic material [44]. One of the two patients reported fever three days before the test date, while the other was completely asymptomatic. Although its virus was viable, quantification cycles were very high: a late diagnosis was more likely than a precocious asymptomatic one, therefore a previous occurrence of overlooked prodromal symptoms could not be ruled out.

A study estimated mpox serial interval and incubation period using a large sample from the UK Health Security Agency surveillance and contact tracing data [45]. Applying two different Bayesian models, an incubation period resulted in 7.6–7.8 days, while a serial interval was set at 8.0–9.5 days. Evaluating the relative times from the symptom onset date in the primary case to the date of exposure for the secondary contact, serial interval, and the incubation period of a small number of case–contact pairs, for whom all events could be linked, the authors found in most cases negative times from primary case onset to secondary contact exposure. Such negative intervals indicated pre-symptomatic transmission. Assuming statistical independence between the serial interval and incubation period, at least 53% of transmission occurred in the pre-symptomatic phase. Therefore, the real issue might be not the existence of asymptomatic carriers, but the shedding of the competent virus before symptom development, allowing for infection spread. The same conclusion has been described in a study from Antwerp, where 25 contacts (either sexual or non-sexual) of confirmed mpox cases were evaluated after the known exposure [46]. Thirteen of them developed some evidence of infection, with viral DNA detected 4 days before symptom onset in five cases. A replication-competent virus was found during the pre-symptomatic phase in three out of these five cases.

Our study suffers from some limitations. Firstly, the small sample size could have hampered the detection of rare events. Although consistent with the planned statistical methods, it might be inadequate to detect events much less common than that reported by published literature. Half of enrolled individuals received some dose of vaccination. Such non-homogeneous features of the study population might have hindered the potency of analyses since immunization impact was not considered for the sample size calculation. Although the rate of vaccination was low, a recent network-based model demonstrated that vaccination coverage of high-risk MSM of 5%, 25%, and 50% could lead to a 56%, 91%, and 95% reduction in the number of cases, respectively [47]. Indeed, recent data from an Italian study exhibited a decrease in mpox cases of 7.4% per week in the post-scale-up of media and public communication campaign (13 June–21August) and of 26.6% per week in the post-vaccination period (22 August–30 November) [48]. This model suggests that vaccination significantly influenced the outbreak in combination with several other factors, such as communication. Thus, the calculation of our sample size should have considered the incidence occurring during the study period and then recalculated because of the delayed protocol start for bureaucratic reasons. Additionally, most case series identified PLWH as the population most involved in the epidemic, but we were able to recruit only a small proportion from this group. Their recruitment through advertisements might have chosen people who were more concerned about contagion, resulting in a selection bias. Secondly, the study was performed during the declining phase of the outbreak with an overall low number of new cases. The results might be by some means different in other time spans during the 2022–2023 outbreak [49]. Thirdly, the risk of acquiring the infection could have been reduced by other factors, such as a possible behavioral change in response to the mpox outbreak, as suggested by an AMIS online survey [50]. In addition, our results could not be representative of the whole national epidemic—although Milan alone represented more than 40% of all Italian cases—or be broadened to other geographic locations. Lastly, we were able to test only pharyngeal and anal samples, so we could not exclude the presence of mpox virus in other anatomical sites, such as skin, urine, conjunctiva, or semen.

## 5. Conclusions

Our findings suggest that the prevalence of asymptomatic mpox carriers is lower than expected. Pre-symptomatic shedding of the replication-competent virus represents a major issue for epidemic containment, while further data based on intensive screening in high-risk populations are needed to better define mpox circulation in asymptomatic people and its role for the control of possible new outbreaks.

## Figures and Tables

**Figure 1 pathogens-12-00798-f001:**
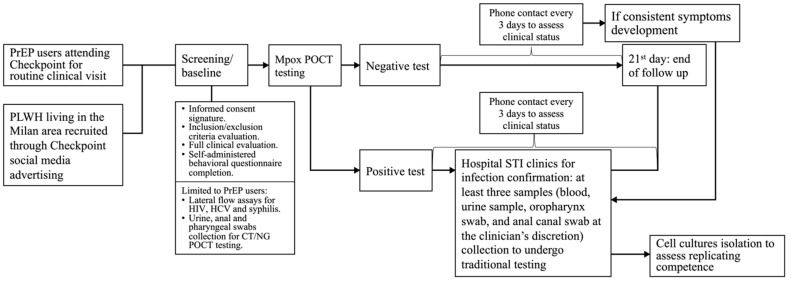
Detailed flow chart of study procedures.

**Figure 2 pathogens-12-00798-f002:**
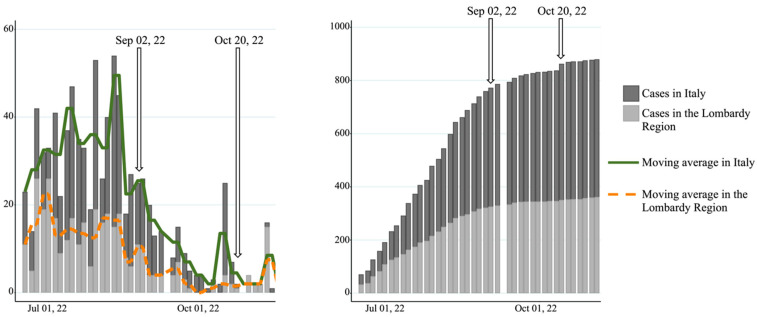
Epidemiologic trend of mpox in the Lombardy region and in Italy from outbreak start-up to study period execution. Discrete three-day count (**left**) and cumulative case counts (**right**) are shown, as well as the weekly moving averages. Based on data retrieved from Ministero della Salute–Mpox Bollettino.

**Table 1 pathogens-12-00798-t001:** Demographic, clinical, and behavioral features of study population.

		Enrolled Subjects (N = 72)
**Age, years, median (IQR)**		37 (32–46)
Enrollment group, n (%)	PLWH	9 (12.5)
	PrEP users	63 (87.5)
Length of PrEP use, months, median (IQR)		25 (17–32)
White ethnicity, n (%)		67 (93.1)
Level of education, n (%)	University degree	54 (75.0)
	Secondary school	16 (22.2)
	Lower level	2 (2.8)
Employed, n (%)		58 (80.6)
Any STI in the previous 3 months, n (%)		10 (13.9)
	Syphilis	1 (1.4)
	Chlamydia	2 (2.8)
	Gonorrhea	5 (6.9)
	HSV-1/2	2 (2.8)
	Anal condylomas	2 (2.8)
Number of sexual intercourses in the previous 3 months, median (IQR)		11 (4–24)
Number of condomless sexual intercourses in the previous 3 months, median (IQR)		3 (1–15)
Use of any recreative drug in the previous 4 weeks, n (%)		19 (26.4)
Chemsex practices (any)^§^, n (%)		4 (5.6)
	Cocaine hydrochloride	6 (8.3)
	Cocaine freebase	2 (2.8)
	MDMA	4 (5.6)
	Crystal meth	3 (4.2)
	Mephedrone	4 (5.6)
	GHB/GBL	3 (4.2)
	THC	9 (12.5)
	Ketamine	4 (5.6)
	Popper	10 (13.9)
	MDPV	4 (5.6)
	Heroine	1 (1.4)
5-Phosphodiesterase inhibitors use, n (%)		12 (16.7)
Concomitant syphilis diagnosis *, n (%)		2 (3.2)
Concomitant gonorrhea diagnosis *, n (%)		2 (3.2)
Concomitant Chlamydia diagnosis *, n (%)		5 (7.9)
Fully vaccinated °, n (%)		20 (27.8)
	No vaccine	30 (41.7)
	Vaccine during childhood	10 (13.9)
	Vaccine during childhood plus one booster	6 (8.3)
	One injection	12 (16.7)
	Two injections	14 (19.4)
Time from first vaccine injection, days, median (IQR)		16 (10–21)
Time from vaccine completion, days, median (IQR)		11 (3–26)

HSV: herpes simplex virus; MDMA: 3,4-methylenedioxy-methamphetamine; GHB/GBL: gamma-hydroxybutyrate/gamma butyrolactone; THC: tetrahydrocannabinol; MDPV: 3,4-Methylenedioxypyrovalerone. § sexualized use of crystal meth, mephedrone, and/or GHB/GBL. * STI testing performed only on PrEP users (N = 88). ° having received the vaccination during childhood plus one booster or having received two injections of MVA-BN in 2022. According to the Italian recommendations, PLWH with full vaccination during childhood should receive only one booster dose.

## Data Availability

All the anonymized data used to perform this analysis will be available for any further revision upon request.

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
