# Peer review of "Detection of Asymptomatic Mpox Carriers among High-Ri Men Who Have Sex with Men: A Prospective Analysis"

_pathogens, 2023, doi:10.3390/pathogens12060798_

Round 1

Reviewer 1 Report

This study describes the prevalence of mpox in MSM PrEP users and PLWH and discussed about the asymptomatic carriers on viral spread. The topic is important since it is closely related to the society. However, this paper got several issues, and the study is not comprehensively designed and discussed. 

1.     In your abstract, you mentioned PrEP users and PLWH. First of all, please use full spellings instead of abbreviations in the abstract. Secondly, please enrich your abstract with slightly more background information.

2.     In your introduction, please describe more about Mpox related vaccines. You mentioned about vaccination in results and discussion; however, you never talk about it in detail. 

3.     In your introduction, please talk about the pathogenesis of Mpox. For example, what are the symptoms? Is there any incubation period already known? You focus on MSM, but is there any data about women and child having Mpox due to MSM? 

4.     In your introduction, please add enough references (see line 30-33, 34-35, 40-44, 45-46, 46-48). Usually, people cite papers after each solid statement. Please go over your paper and add references accordingly. This suggestion is not limited to introduction. 

5.     For methods, it would be great to summarize your steps in a figure. Something like that you put each step in a box and use arrows to connect the boxes, which provide an extra illustration to show readers about the methods directly.

6.     You concluded that the prevalence of asymptomatic mpox carriers is lower than expected. What is your previous expectation based on existing data? Any asymptomatic data on mpox related viruses that is known? These points should be included in introduction and discussion. What are the factors related to asymptomatic infection in mpox or mpox related viruses that has been reported? An example is incubation period, as mentioned in comment 3. Previous reports indicate that mpox incubation period is 3-17 days, which is a time span that could be tricky. Even incubation period itself, one factor, could impact the entire study. Also, in your conclusion, you mentioned only self-monitoring. What is self-monitoring, and how could self-monitoring be done? That’s a very tricky thing to say. Is there any data about successful or unsuccessful cases regarding to this? The conclusion has a bias and is misleading.

The quality of English is relatively ok, may require minor English editing.

Author Response

  • In your abstract, you mentioned PrEP users and PLWH. First of all, please use full spellings instead of abbreviations in the abstract. Secondly, please enrich your abstract with slightly more background information -> abstract has been fully re-written to better comply with both reviewers’ requests. We hope that in this edited format could be suitable for publication.
  • In your introduction, please describe more about Mpox related vaccines. You mentioned about vaccination in results and discussion; however, you never talk about it in detail. -> a paragraph has been added in the “Introduction” section to add some information as requested.
  • In your introduction, please talk about the pathogenesis of Mpox. For example, what are the symptoms? Is there any incubation period already known? You focus on MSM, but is there any data about women and child having Mpox due to MSM?-> these topics go beyond what was intended for our research, nevertheless a paragraph has been added to include information about the topics requested by the Reviewer.
  • In your introduction, please add enough references (see line 30-33, 34-35, 40-44, 45-46, 46-48). Usually, people cite papers after each solid statement. Please go over your paper and add references accordingly. This suggestion is not limited to introduction.-> more references have been added.
  • For methods, it would be great to summarize your steps in a figure. Something like that you put each step in a box and use arrows to connect the boxes, which provide an extra illustration to show readers about the methods directly. -> another figure has been added to clarify study procedures.
  • You concluded that the prevalence of asymptomatic mpox carriers is lower than expected. What is your previous expectation based on existing data? Any asymptomatic data on mpox related viruses that is known? These points should be included in introduction and discussion. What are the factors related to asymptomatic infection in mpox or mpox related viruses that has been reported? -> our study is based on two reports (De Baetselier I, et al., Nat Med 2022; Ferré VM, et al., Ann Intern Med 2022) that are clearly cited in the “Introduction” section (see ref. 25,26). All matters that were so far known about asymptomatic infections come from these papers.Some additional features are present in the studies cited in the “Discussion” section (see ref. 36,37).
  • An example is incubation period, as mentioned in comment 3. Previous reports indicate that mpox incubation period is 3-17 days, which is a time span that could be tricky. Even incubation period itself, one factor, could impact the entire study. -> as previously requested, we added information about incubation period that was traditionally set at 5-21 days, although in the 2022-23 outbreak seems considerably shorter (7 days). To overcome this issue, we required in the inclusion criteria the absence of consistent symptoms in the previous 21 days and the follow up after testing has a length of 21 days (for a global 42-day span of potential disease monitoring). We agree that incubation period is a problem with published literature: the mentioned papers by De Baetselier and Ferré, but also by Pestel et colleagues (which has an approach similar to ours) lack this issue. We think that our study design — that complies with the longest accepted incubation interval (21 days) — could reasonably overcome such limitation.
  • Also, in your conclusion, you mentioned only self-monitoring. What is self-monitoring, and how could self-monitoring be done? That’s a very tricky thing to say. Is there any data about successful or unsuccessful cases regarding to this? The conclusion has a bias and is misleading -> self-monitoring is recommended for contacts of probable and confirmed cases by the WHO (see https://www.who.int/publications/i/item/WHO-MPX-Surveillance-2022.4) and by the ECDC (see https://www.ecdc.europa.eu/sites/default/files/documents/Public%20health%20considerations%20for%20mpox%20in%20EUEEA%20countries%202023_0.pdf). Since this approach is recommended by international Health Authorities, we do not agree that our conclusions might be misleading while in line with these recommendations. Anyway, references have been added to clarify this issue.

Reviewer 2 Report

Thank you for counting on me for this revision. 

Please, take into account my considerations 

The concerns raised by this manuscript are briefly reported by the authors in the discussion.  However, they are methodological limitations determinant for the validity of the study.

The population chosen is not homogeneous for the purposes of the study. It is therefore not valid for drawing the conclusions offered. 

Although there are no studies of actual efficacy of the smallpox vaccine against mpox, the studies on the basis of which its use has been recommended suggest that the vaccinated versus unvaccinated population is not comparable for acquiring infection or for being asymptomatic carriers of the virus. 

This idea is also supported by their reference 11. Among contacts of smallpox patients, Sark found a higher proportion of positivity for smallpox virus detection among unvaccinated than among vaccinated asymptomatic contacts.

The sample size calculation should take into account the population prevalence of the disease at the time of the study. 

Both, acquiring this transmissible disease, and being an asymptomatic carrier, are directly related to the local prevalence figures. 

The authors' graph shows that at the time of data collection, the prevalence was very low. It is possible that the sample size is much smaller than necessary. Even more so if it is decided to separate the vaccinated and unvaccinated groups. 

It seems that there is a selection bias, since the method chosen for recruitment through advertisements may favour people who are more concerned and careful to avoid contagion. In fact, the percentage of condom use is much higher than in other series. 

Regarding the anatomical location where samples were taken from each participant. There is evidence that positive PCRs are more frequent and more prolonged in cutaneous-mucosal lesions (https://doi.org/10.1016/S1473-3099(22)00794-0). In asymptomatic patients, it would be appropriate to have analysed the swabs from the skin areas, even without lesions, where lesions have been most frequently described in the mpox outbreak.

It is necessary to know how the following exclusion criteria were assessed:

iii) no skin lesions of novel appearance in the previous 21 days; 

(iv) no clinically evident symptoms present during a routine evaluation held by an infectious diseases specialist or by a trained peer;

Skin lesions are very often found in locations that are difficult to access visually and in most cases, in immunocompetent persons, are small and few in number. They may therefore be overlooked if a detailed cutaneous-mucosal examination has not been done. The report should specify how this evaluation was done. 

(iii) is it self-reported?

(iv) Was a detailed cutaneous-mucosal examination including poorly accessible areas (genito-anal and oral mucosa among them) part of this assessment?

Finally, self-sampling for mpox is not validated in clinical practice, so the authors should refer to a study that confirms its validity. I am only aware of this reference to self-collection in MPOX doi: 10.1093/cid/ciac889. PMID: 36370091. 

Regarding wording:

-The rapid Standard M10 platform used was certified to test oral samples only, and reference 7 do not use this technique, so the use of this particular platform for samples other than oral samples is not supported by the reference. The wording is misleading. 

As a summary, this paper, after clarifying the points requested, can reflect on the title, provide conclusions, and talk in the discussion about:

-Detection of asymptomatic MonkeyPox virus carriers

-by pharyngeal and anal sampling

-analysed with the rapid Standard M10 platform 

-during the decline in mpox population prevalence

-in a cohort of  N people attended  in a  community-based facility for PrEP users. 

Author Response

  • The concerns raised by this manuscript are briefly reported by the authors in the discussion. However, they are methodological limitations determinant for the validity of the study. The population chosen is not homogeneous for the purposes of the study. It is therefore not valid for drawing the conclusions offered. -> we do not agree about this concern since a not homogeneous population was an advantage of our research and not a limitation. The mentioned papers by De Baetselier (Nat Med 2022) and Ferré (Ann Intern Med 2022) retrospectively selected all individuals attending their STI clinics without specifically focusing on groups at major risks. Epidemiological data show that half of cases are registered in PLWH and a quarter in PrEP users, thus our study population represents a group at considerable risk of exposure to mpox, as might be inferred by the high screening failure rate due to previous mpox diagnosis. Ferré found a prevalence of 6.5% in the general MSM community attending a STI clinic after excluding those who had a concomitant Chlamydia or gonorrhea infection: since we chose a study population at higher risk (PrEP, PLWH, not excluding the presence of other STIs) we expected to find at least a comparable proportion of asymptomatic infections. Anyway, to clarify this topic a couple of sentences have been added in the “Introduction” and “Methods” sections.
  • Although there are no studies of actual efficacy of the smallpox vaccine against mpox, the studies on the basis of which its use has been recommended suggest that the vaccinated versus unvaccinated population is not comparable for acquiring infection or for being asymptomatic carriers of the virus. This idea is also supported by their reference 11. Among contacts of smallpox patients, Sark found a higher proportion of positivity for smallpox virus detection among unvaccinated than among vaccinated asymptomatic contacts. -> we do agree that vaccination significantly reduces the risk of disease acquisition, and we acknowledged that in the Limitation paragraph of our manuscript. We added some sentences about vaccination efficacy (and failures) to comply with what requested by Reviewer #1. Indeed, in Milan few vaccine doses were available and only a minority of study population received full immunization.
  • The sample size calculation should take into account the population prevalence of the disease at the time of the study. Both, acquiring this transmissible disease, and being an asymptomatic carrier, are directly related to the local prevalence figures. The authors' graph shows that at the time of data collection, the prevalence was very low. It is possible that the sample size is much smaller than necessary. Even more so if it is decided to separate the vaccinated and unvaccinated groups -> we do agree with this point and we acknowledged that in the Limitation paragraph of our manuscript. Although this issue might be intriguing, it was not possible at time of study design (July-August 2022) to forecast what incidence and prevalence would be one month and a half later. Calculating an exact and punctual sample size could be anyway tricky given the absence of the denominator (number of sexually active non strictly monogamous MSM who had not acquired the infection at the beginning of September 2022). We based our calculation on the available published data Ferré (Ann Intern Med 2022). Interestingly, the study by Ferré was performed from June 2022, when the curve was at the beginning of its rise, while the study by De Baetselier was performed in May, when cases were still sporadic in Belgium. Additionally, several reports indicate a failure of “one-shot only” or recent full vaccination (mentioned in our manuscript). Thus, we do not think sample size calculation should be re-run. Nevertheless, the weight of this limitation has been further underlined within the dedicated paragraph.
  • It seems that there is a selection bias, since the method chosen for recruitment through advertisements may favour people who are more concerned and careful to avoid contagion. In fact, the percentage of condom use is much higher than in other series.-> we do agree with this point but recruitment through advertisements accounts only for PLWH (9.3% of study population), while the others have been consecutively enrolled during routine clinical evaluation for PrEP. Condom effectiveness against mpox has not a clear role but seems limited (see Strathdee, Lancet Infect Dis 2023). Anyway, this issue has been added as a further limitation in the dedicated paragraph.
  • Regarding the anatomical location where samples were taken from each participant. There is evidence that positive PCRs are more frequent and more prolonged in cutaneous-mucosal lesions (https://doi.org/10.1016/S1473-3099(22)00794-0). In asymptomatic patients, it would be appropriate to have analysed the swabs from the skin areas, even without lesions, where lesions have been most frequently described in the mpox outbreak. -> we do agree that quantification cycles are generally very high when swabs are taken from skin lesions. However, in absence of evident lesions, it would be hard to standardize a site of collection on the skin. The paper mentioned by Reviewer #2 refers to analyses on individuals with well symptomatic clinical manifestations. The POCT used for mpox testing is registered only for pharyngeal samples, we decided to add a second swab with anal material after the publication of the paper by Tarín-Vicente et colleagues (Lancet 2022) where quantification cycles were more significantly related to anal canal especially in case of receptive intercourses even if the rapid Standard M10 platform was not registered for this anatomical site. Although it would be of great interest to assess also random skin samples, urine, or semen materials, they would not be suitable for testing. This issue has been added to the Limitation paragraph.
  • It is necessary to know how the following exclusion criteria were assessed: (iii) no skin lesions of novel appearance in the previous 21 days; (iv) no clinically evident symptoms present during a routine evaluation held by an infectious diseases specialist or by a trained peer. Skin lesions are very often found in locations that are difficult to access visually and, in most cases, in immunocompetent persons, are small and few in number. They may therefore be overlooked if a detailed cutaneous-mucosal examination has not been done. The report should specify how this evaluation was done. (iii) is it self-reported? (iv) Was a detailed cutaneous-mucosal examination including poorly accessible areas (genito-anal and oral mucosa among them) part of this assessment? -> we would thank Reviewer #2 for giving us the chance to further specify how the clinical evaluation has been performed on the screening day: the accurate clinical evaluation was the hardest part of our work since it was very time-consuming. Some sentences have been added to clarify this point. Concerning the medical history of the previous 21 days, of course they were self-reported. Screening procedure was detailed, and questions were focused on symptoms that might be related to other illnesses, thus restricting individual selection (i.e., we did not ask “had you fever because of mpox in the previous 21 days?” but “have you recorded any episode of fever of any degree in the previous 21 days?”). Although reporting bias could not be completely ruled out, we think that this issue has minimal significance to our study since no positive test was found.
  • Finally, self-sampling for mpox is not validated in clinical practice, so the authors should refer to a study that confirms its validity. I am only aware of this reference to self-collection in MPOX doi: 10.1093/cid/ciac889. PMID: 36370091. -> self-collection of samples for STI testing is an established procedure that resulted reliable and well-accepted through many studies; thus, it is a standard procedure in Milano Checkpoint ETS. The proposed reference has been added for completeness.
  • Regarding wording: -The rapid Standard M10 platform used was certified to test oral samples only, and reference 7 do not use this technique, so the use of this particular platform for samples other than oral samples is not supported by the reference. The wording is misleading.-> we rewrote the paragraph to clarify this item.
  • As a summary, this paper, after clarifying the points requested, can reflect on the title, provide conclusions, and talk in the discussion about: -Detection of asymptomatic MonkeyPox virus carriers; by pharyngeal and anal sampling; -analysed with the rapid Standard M10 platform; -during the decline in mpox population prevalence; -in a cohort of N people attended in a community-based facility for PrEP users. -> editing has been performed to comply with this comment.

Round 2

Reviewer 1 Report

1.     As mentioned in the last peer-review report, for abstract, please use full spellings instead of directly using abbreviations. Enriching abstract does not mean you put method details there, for instance, you don’t have to put the inclusion criteria, and how many subjects you screened. You can give a brief intro on Mpox, like what are already known symptoms, transmission pathways and the known impacts to human society, then what’s the question that haven’t been fully clarified (which is your topic), then a short summary on what you have done and what you have found.   

2.     The introduction part is enriched. However, things should be described in a more ordered way. Currently, you talked about the number of cases; then the groups that the study is focused on; then vaccines and drugs; then symptoms, incubation period, and transmissions other than what you are focusing on; then your original introduction parts on MSM and related stuff. The bullet points are important things to talk about in an introduction. However, please try to combine viral things together first, and then introduce on the specific topic you are focusing on and related stuff. In this way, the introduction part will have a more logical flow. 

3.     For figure1, too much information is listed. Please try to make things more direct and clearer. There’s also a box with smaller fonts. Please re-organize your figure 1. 

4.     For figure 2, please indicate and cite where did you obtained the epidemiologic trend of mpox in Lombardy Region. 

5.     In addition, as you mentioned in the manuscript, your study has a limited sample size from a specific location. So, can your result be a reference to other places in Italy, in Europe, or any other regions around the world based on any known data? That’s something else you can put in your discussion.

6.     For your conclusion, you said self-monitoring, which is recommended by “Health Authorities”. The current way you conclude your paper could be misleading, since you never really talk about what the “health authorities” recommend. What are the details, like are there any prerequisites on any statements of their recommendations? You can make a different paragraph in your discussion to support your conclusion. 

Some long sentences may not be clear to readers and will need editors to double check. Overall the quality of English seems all right. 

Author Response

  • As mentioned in the last peer-review report, for abstract, please use full spellings instead of directly using abbreviations. Enriching abstract does not mean you put method details there, for instance, you don’t have to put the inclusion criteria, and how many subjects you screened. You can give a brief intro on Mpox, like what are already known symptoms, transmission pathways and the known impacts to human society, then what’s the question that haven’t been fully clarified (which is your topic), then a short summary on what you have done and what you have found.-> this request was somehow difficult to satisfy since generally abstract should be concise and could not contain too many background details. Additionally, we do believe that the importance of our data relies on the accuracy of study design: reducing the description of study procedures could impoverish the meaning of our work. Nevertheless, we tried to comply with the observations by Reviewer #1 remaining within the required low word count.
  • The introduction part is enriched. However, things should be described in a more ordered way. Currently, you talked about the number of cases; then the groups that the study is focused on; then vaccines and drugs; then symptoms, incubation period, and transmissions other than what you are focusing on; then your original introduction parts on MSM and related stuff. The bullet points are important things to talk about in an introduction. However, please try to combine viral things together first, and then introduce on the specific topic you are focusing on and related stuff. In this way, the introduction part will have a more logical flow. -> the “Introduction” section has been re-shaped. We hope that now could appear more logical and clearer.
  • For figure1, too much information is listed. Please try to make things more direct and clearer. There’s also a box with smaller fonts. Please re-organize your figure 1. -> this Figure has been designed according to the requests by Reviewer #2 who approved it in its actual shape. We eliminated the box with smaller fonts that might be non-essential for understanding study procedures.
  • For figure 2, please indicate and cite where did you obtained the epidemiologic trend of mpox in Lombardy Region. -> a note has been added.
  • In addition, as you mentioned in the manuscript, your study has a limited sample size from a specific location. So, can your result be a reference to other places in Italy, in Europe, or any other regions around the world based on any known data? That’s something else you can put in your discussion. -> Milan alone represents more than 40% of all Italian cases, as has been underlined: nevertheless, our data might not be generalized to other locations. Another point in the limitation paragraph has been added to address this issue.
  • For your conclusion, you said self-monitoring, which is recommended by “Health Authorities”. The current way you conclude your paper could be misleading, since you never really talk about what the “health authorities” recommend. What are the details, like are there any prerequisites on any statements of their recommendations? You can make a different paragraph in your discussion to support your conclusion. -> this sentence has been added upon request by Reviewer #2. We tried to clarify the point in the Introduction section so that this conclusion would not be perceived as “misleading”.

Reviewer 2 Report

Many thanks to the authors for their responses to my comments, which were only intended to help improve the manuscript.

 Please note these minor corrections to the new text 

The toxicity profile of Cidofovir's pro-drug Brincidofovir, and Cidofovir is very different. If the text addresses this issue, I recommend modifying the sentence that seems to make them the same in this respect. 

In addition to this sentence: 

Thirdly, the risk of acquiring the infection could have been reduced by other factors: half of study population received some dose of vaccination;

Should be added in limitations the fact that half of the participants had received some dose of the vaccine makes the population non-homogeneous. However, the study did not find any positive cases, so we believe that this design drawback does not hinder the assessment of the results. 

Tiny type in abstract 

We aimed to prospectively assess the presence of mpox in asymptomatic high-risk MSM using HIV pre-exposure prophylaxis (PrEP) users (remove users or using)

Further work by the same Belgian team discussed in this manuscript supports the authors’ conclusions. They don’t found asymptomatic cases among high risk population considering the updated mpox case definition. It is important to add the comment and cite the paper in this manuscript.

Author Response

  • The toxicity profile of Cidofovir's pro-drug Brincidofovir, and Cidofovir is very different. If the text addresses this issue, I recommend modifying the sentence that seems to make them the same in this respect. -> the sentence has been changed.
  • In addition to this sentence: Thirdly, the risk of acquiring the infection could have been reduced by other factors: half of study population received some dose of vaccination. Should be added in limitations the fact that half of the participants had received some dose of the vaccine makes the population non-homogeneous. However, the study did not find any positive cases, so we believe that this design drawback does not hinder the assessment of the results. -> the limitation paragraph has been changed to underline this feature.
  • Tiny type in abstract. We aimed to prospectively assess the presence of mpox in asymptomatic high-risk MSM using HIV pre-exposure prophylaxis (PrEP) users (remove users or using) -> edited.

Round 3

Reviewer 1 Report

1. For your figure 1, still a lot of information, but at least make the font consistent.

2. For your figure 2, "...available at..", please make a formal citation even though you are citing something from a website. 

English seems all right. Editors may need to do minor changes.

Author Response

Following a point-by-point responses to the issues:

1. For your figure 1, still a lot of information, but at least make the font consistent. -> we are glad that in this new format might be suitable to be published.

2. For your figure 2, "...available at..", please make a formal citation even though you are citing something from a website. -> a formal citation has been added as suggested.